# Diversity of Geophilic Dermatophytes Species in the Soils of Iran; The Significant Preponderance of *Nannizzia fulva*

**DOI:** 10.3390/jof7050345

**Published:** 2021-04-28

**Authors:** Simin Taghipour, Mahdi Abastabar, Fahimeh Piri, Elham Aboualigalehdari, Mohammad Reza Jabbari, Hossein Zarrinfar, Sadegh Nouripour-Sisakht, Rasoul Mohammadi, Bahram Ahmadi, Saham Ansari, Farzad Katiraee, Farhad Niknejad, Mojtaba Didehdar, Mehdi Nazeri, Koichi Makimura, Ali Rezaei-Matehkolaei

**Affiliations:** 1Department of Medical Parasitology and Mycology, Faculty of Medicine, Shahrekord University of Medical Sciences, Shahrekord 88157-13471, Iran; simintaghipoor@yahoo.com; 2Invasive Fungi Research Center, Department of Medical Mycology and Parasitology, School of Medicine, Mazandaran University of Medical Sciences, Sari 48157-33971, Iran; mabastabar@gmail.com (M.A.); mjpezeshk2@gmail.com (M.R.J.); 3Infectious and Tropical Diseases Research Center, Health Research Institute, Ahvaz Jundishapur University of Medical Sciences, Ahvaz 61357-15794, Iran; piri.fahimeh@gmail.com; 4Department of Medical Mycology, School of Medicine, Ahvaz Jundishapur University of Medical Sciences, Ahvaz 61357-15794, Iran; aboualielham@gmail.com; 5Allergy Research Center, Mashhad University of Medical Sciences, Mashhad 91766-99199, Iran; h.zarrin@gmail.com; 6Medicinal Plants Research Center, Yasuj University of Medical Sciences, Yasuj 75919-94799, Iran; nooripoor8561@gmail.com; 7Department of Medical Parasitology and Mycology, School of Medicine, Infectious Diseases and Tropical Medicine Research Center, Isfahan University of Medical Sciences, Isfahan 81746-73461, Iran; dr.rasoul_mohammadi@yahoo.com; 8Department of Medical Laboratory Sciences, Faculty of Paramedical, Bushehr University of Medical Sciences, Bushehr 75187-59577, Iran; bahram.ahmadi.fahlyan@gmail.com; 9Department of Medical Parasitology and Mycology, School of Medicine, Shahid Beheshti University of Medical Sciences, Tehran 19857-17443, Iran; samansari1988@yahoo.com; 10Department of Pathobiology, Faculty of Veterinary Medicine, University of Tabriz, Tabriz 51666-16471, Iran; katiraee_f@yahoo.com; 11Laboratory Sciences Research Center, Golestan University of Medical Sciences, Gorgan 49189-36316, Iran; fniknezhad@yahoo.com; 12Department of Medical Mycology and Parasitology, School of Medicine, Arak University of Medical Sciences, Arak 63417-38481, Iran; didehdar_m@yahoo.com; 13Infectious Diseases Research Center, Kashan University of Medical Sciences, Kashan 87159-73474, Iran; mehdinazeri@yahoo.com; 14Laboratory of Medical Mycology, Graduate School of Medicine, Teikyo University, Tokyo 173-8605, Japan; makimura@med.teikyo-u.ac.jp

**Keywords:** geophilic dermatophytes, *Nannizzia fulva*, *Arthroderma*, ITS sequencing, Iran

## Abstract

A molecular epidemiology study was conducted between 2016 and 2017 by a network of collaborators from 12 provinces in the Islamic Republic of Iran. A total of 1484 soil samples from different habitats were screened for the presence of dermatophytes by using the hair baiting technique. The primary identification of isolates was carried out by amplification and MvaI restriction fragment length polymorphism (RFLP) of the internal transcribed spacers regions of ribosomal DNA (ITS-rDNA). The identifications, especially in the cases of isolates with unknown RFLP patterns, were confirmed by sequencing of the ITS-rDNA region. As a result, 256 isolates were recovered. The isolation rate was higher in soils with pH range 7.1–8.0, collected from animal habitats (*n* = 78; 34%) and parks and gardens (*n* = 75; 32%), geographically from Mazandaran Province (*n* = 115; 49.5%) and seasonally in the spring (*n* = 129; 50.4%), all of which were statistically significant (*p* < 0.05). The dermatophytes comprising five species of the two genera, viz., *Nannizzia fulva* (*n* = 214), *N. gypsea* (*n* = 34), *Arthroderma quadrifidum* (*n* = 5), *A. gertleri* (*n* = 2) and *A. tuberculatum* (*n* = 1), were isolated. The geophilic dermatophytes occurred in various soils from different parts of Iran; however, surprisingly, *N. fulva* emerged as the dominant species, outnumbering the common geophilic species of *N. gypsea*. For the definitive identification of soil inhabitant dermatophytes, DNA-based identification is strongly recommended.

## 1. Introduction

It is known that soil is a possible reservoir of some fungal pathogens, causing cutaneous infections in humans and animals, among which dermatophytes are the most important [1,2]. Dermatophytes are a group of filamentous fungi and encompass the seven genera of *Trichophyton*, *Microsporum*, *Epidermophyton*, *Nannizzia*, *Arthroderma*, *Lophophyton* and *Paraphyton* [3]. The dermatophytes species are of veterinary and public health significance, because they can invade the stratum corneum of the skin and its appendages such as nails and hair in both humans and animals, causing infections medically termed as dermatophytosis (ringworm) [4]. Ecologically, most dermatophyte species are anthropophilic (human-adapted) or zoophilic (related to animal dwellings), while the third group (geophilic) resides in soils and are termed geophilic. The occurrence of infections by geophilic dermatophytes is low but continuous, and their ability to cause human and animal infections is also well-known around the world, thus drawing the attention of medical and veterinary mycologists [5,6]. In public places such as parks and gardens and, also, in animal residences, the soil is continuously manipulated by humans and animals. Then, it is logical to imagine that organic keratinous debris are constantly mixed with the soil, and that such soils, if contaminated with pathogenic keratinophilic fungi, may infect humans and animals [7]. The innovation of the hair bait technique by Vanbreuseghem [8] in 1952, on the one hand, and on the other hand, the application of molecular approaches have increased our understanding of the diversity and ecology of soil fungi [9,10]. A study on the occurrence of keratinophilic fungi, including dermatophytes in the soils of Iran, was launched in 2002 by Shadzi et al. in Isfahan [11]. Since then, few investigations have been performed on the dermatophytes mycoflora of the soil [10,12,13,14], but the diversity of dermatophyte species in soils from most parts of the country remains largely unknown. The Islamic Republic of Iran, commonly known as Iran, is geographically located in West Asia. It is the second-largest country in the Middle East and 17th largest in the world, covering 636,372 square miles. The country is characterized by 11 of the 13 world’s climates, ranging from arid and semi-arid to a subtropical climate, and has four distinct seasons spread throughout the year. However, not all parts of the country experience all four seasons: (What Type of Climate Does Iran Have? Accessed 20 June 2019, <https://www.worldatlas.com/articles/what-type-of-climate-does-iran-have.html>). In this study, by using sequence-based methods of PCR-RFLP and PCR sequencing, we aimed to characterize the species composition and distribution profile of geophilic dermatophytes in soils from 12 different provinces of Iran, with respect to the seasonal status and ecological niche.

## 2. Methods

### 2.1. Locations and Selection of Sites for Collection of Soil Samples

A total of 1484 soil samples were collected during November 2016 to the end of September 2017 from different habitats in 12 governorates of Iran. The sampling sites were selected on the basis of the likely presence of soil with keratin residues from humans and animals, e.g., garden and park, mountain, animal habitat, roadside, home range, riverside and schools. A small amount of soil sample (10 g) was transferred to a 50-mL falcon tube containing 100-mL double-distilled water (ddw), and the mixture was shortly agitated then allowed to stand for about 30 min. A pH electrode (Knick Portamess^®^ 911 pH meter, Berlin, Germany) was inserted into the solution, and the acidity was read.

### 2.2. Fungal Isolation and Purification

Around 100–200 g of soil from the superficial layer at a depth not exceeding 5 cm was picked up with a plastic disposable spoon and placed in a single-use plastic bag. For fungal isolation by the Vanbreuseghem technique [8], a sterile Petri dish was filled with the soil sample; then, fragments of sterilized human (girl) hairs were sprinkled over the soil for baiting. The hair-baited soil dishes were moistened with sterile distilled water supplemented with 0.5-mg/mL cycloheximide (Sigma-Aldrich Co, Ltd., St. Louis, MO, USA) and 5-mg/mL chloramphenicol (Sigma-Aldrich Co, Ltd., St. Louis, MO, USA) incubated at 28 °C and checked daily for the fungal growth for up to 8 weeks. Fungal growths appearing on baited hairs were stained with lactophenol aniline blue solution and examined microscopically. In the case of the presence of fungal elements characteristic for a dermatophyte growth, the invaded hairs were inoculated with Mycosel agar (BD Diagnostics, Becton Drive, Franklin Lakes, NJ, USA) to get a pure culture. Each grown colony was microscopically checked, and the pure isolate was preliminarily recognized by phenotypic characteristics at the genus/species level.

### 2.3. Molecular Identification

In this study, for preliminary molecular screening/identification of the isolates, we used amplification and the *Mva*I restriction fragment length polymorphism (RFLP) of the internal transcribed spacer (ITS) regions of the rDNA (ITS-rDNA). Briefly, the DNA was mechanically extracted by using the method described previously [10]. Then, amplification of the ITS-rDNA regions was accomplished by using the primer pair ITS1 and ITS4 [15]. The amplified products were then subjected to digestion with a *Mva*I restriction enzyme following the manufacture’s instruction (Thermo Fisher Scientific, Waltham, MA, USA). The fractionized products were separated through agarose gel (2%) electrophoresis, and each isolate was identified on the species level via a size comparison of the obtained bands with those reported in a previous study [16].

### 2.4. Sequencing

To corroborate the identification made by the ITS-RFLP findings, and also, to distinguish some isolates whose RFLP patterns were unknown, 86 isolates from culture-positive samples were subjected to sequencing of the ITS r-DNA regions as a gold standard. Briefly, the ITS rDNA regions were amplified and sequenced with the ITS1/ITS4 primer pair [15] in an ABI Prism™ 3730 genetic analyzer (Applied Biosystems, Foster City, CA, USA). The obtained sequences were then edited and blasted against known sequences in the validated Dermatophyte Database of the Westerdijk Fungal Biodiversity Institute (Utrecht, The Netherlands) to provide species identification. All generated sequences in the study were submitted to GenBank.

### 2.5. Statistical Analysis

The effects of the variables such as soil habitat, location of isolation, soil acidity (pH) and season of sampling on the isolation rate and type of isolated species were statistically examined using the chi-square (x^2^) test with SPSS software version 21 (IBM, Armonk, NY, USA).

## 3. Results

### 3.1. Number of Positive Soil Samples Regarding to Soil pH, Habitat, Geography and Season

In total, in 256 (17.3%) cases, a dermatophyte isolate was recovered from the soil samples. In Table 1 and Figure 1, the frequencies of the isolates regarding different soil habitats, geographic locations, seasons and soil pH were illustrated. According to Table 1, the best isolation rate was accomplished with the soils of animal habitats (34%), parks and gardens (32%). Seasonally, the highest isolation rate (*n* = 129; 50.4%) was achieved in the spring and, geographically, from Mazandaran (*n* = 115; 44.9%) Province. Looking at the soil pH, the isolation rate of the dermatophytes significantly differed, and most of the isolates were recovered from soils with the acidity range 7.1–8.0 (*p* < 0.05). Likewise, the soils from Mazandaran and Khuzestan had, respectively, the highest (50.7%) and the lowest (6.5%) positivity rates of isolation, which were statistically meaningful (*p* < 0.05).

### 3.2. Molecular Identification of Isolates

In Table 2 and Figure 2, the results of the molecular identification are summarized. The amplification of ITS-rDNA in all isolates yielded products ranging from 652 to 677 bp in size. In the primary screening of isolates by the ITS-RFLP profiles, 214 (83.6%) and 34 (13.3%) isolates were, respectively, identified as *N. fulva* and *N. gypsea*, whereas eight isolates created new and unknown RFLP patterns (Figure 2). The sequencing of the representative isolates confirmed the identification of *N. gypsea* and *N. fulva* isolates and revealed the identity of eight unknown strains as *A. quadrifidum* (*n* = 5), *A. gertleri* (*n* = 2) and *A. tuberculatum* (*n* = 1). *Nannizzia fulva* was the predominant species isolated from all the provinces. The new sequences generated in this study were deposited in GenBank (Table 2). The results of similar studies from different countries are summarized in Table 3.

## 4. Discussion

Compared to some previous reports from Iran, narrating the narrow diversity of dermatophytes species in the soils [10,12,13,14], in this assessment, the spectrum of geophilic dermatophytes recovered from the soils has extended to five species, including three new species that have not been reported yet. In our recent study from Khuzestan, southwest of Iran, we hypothesized that *N. fulva* is most likely the main dermatophyte resident in the soils of Iran, and the application of sequence-based methods will clarify this issue [10]. The extensive isolation of *N. fulva* from the soils of 11 additional provinces in the current study confirmed our hypothesis and highlighted the fact that many geophilic soil/clinical isolates formerly reported as *N. gypsea*, on the sole basis of morphological criteria, may actually be other morphological closely related species. The best and the main explanation we have for why *N. fulva* was not reported in the earlier surveys of geophilic dermatophytes in Iran is that the species may has often been misidentified as *N. gypsea*. From the classical circumscriptions of the *N. gypsea* (formerly *Microsporum gypseum*) complex, it is impossible to distinguish *N. gypsea*, *N. fulva* and *N. incurvata* species solely based on their morphological features. Currently, the best strategy to discriminate these taxa is sequence-based methods, i.e., ITS-rDNA RFLP and sequencing [10], as confirmed by the present results. However, the ITS-rDNA restriction banding patterns are not characteristic for all dermatophytes, especially when some species have similar or undescribed restriction profiles. In consensus with this, and as inferred from Figure 2, the three new species detected in this national investigation, i.e., *A. tuberculatum*, *A. gertleri* and *A. quadrifidum*, produced ITS-rDNA restriction profiles that were unknown so far. Therefore, it is a matter of debate whether *N. fulva* and the other, less frequent species isolated in this study are truly rare or misidentified, and therefore, their true incidence is underestimated. A few investigations on geophilic dermatophytes that have attempted to discriminate between members of the *N. gypsea* complex have also supported this issue [12,17,19,20]. In the study of Sharma et al. (2008) on dermatophytes isolated from the soil of an area in Central India by ITS-RFLP and sequencing, 73% of isolates were identified as *N. persicolor*, followed by *N. fulva* (20%) and *N. gypsea* (7%) [17]. That finding was achieved even though *N. persicolor* had never been recorded in India until then. In the survey of keratinophilic fungi in soils of St. Kitts and Nevis, Gugnani et al. using morphological methods found a significant percentage of *N. fulva* (46%), in addition to *N. gypsea* (54%) [19]. Taha et al., in a sequence-based survey on different places in the Sharkia Governorate, Egypt, found *T. mentagrophytes, N. fulva, T. benhamiae* and *A. multifidum*, along with *N. gypsea* (the dominant species), as the spectrum of soil-inhabitant dermatophytes [20]. In Iran, three similar screenings have recently been carried out, all of which were based on the molecular approaches [10,12,14]. In investigations from Ahvaz, southwest, and Isfahan, in the center of Iran, *N. fulva* was the only recovered species, and all the soil isolates morphologically identified as *N. gypsea* were indeed *N. fulva* according to ITS sequencing [10,12]. Soil sampling from the cities in the current study led to a similar finding. However, Pakshir et al. found both *N. gypsea* (80%) and *N. fulva* (20%) among the keratinophilic fungi isolated from soils in Shiraz City, Fars Province, southwest of Iran [14], while all isolates from Fars Province in this study were *N. fulva* and from a city (Noorabad) other than Shiraz. This means that the species arrangement of dermatophytes in the soils of the locations within a province can be different. While members of the *N. gypsea* complex were the dominant species in the soils from different localities in Iran, species other than this complex were more distinguished in the soils from some European countries. In five different reports from 1992 to 2013, *T. ajelloi* (currently, *A. uncinatum*) was the most commonly isolated species from soils in Italy, Poland, Austria and Slovakia [21,22,23,24,25]. Such differences in species domination can be justified with differences in the types of soils sampled. In our study, the majority of isolates were recovered from the soils of animal habitats (34%) and parks and gardens (32%) (which were normally inhabited by cats and dogs). This finding is consistent with the fact that *N. gypsea* and *N. fulva* found universally humus-rich soils or soils inhabited by heavy animal populations that provided a remarkable content of organic matter for these keratinophilic fungi [25,26]. The low occurrence of *A. quadrifidum*, *A. gertleri*, *A. tuberculatum* and other species of the genus *Arthroderma* in previous reports may be due to the misidentification of other non-dermatophyte species. However, in this investigation, all the isolates were identified to the species level by sequence-based procedures, and thus, the infrequency of the mentioned species can possibly be attributed to the fact that these species are weaker soil competitors than the species of the genera *Nannizzia*, *Microsporum* and *Trichophyton*.

A review of the literature indicates that investigations on soil-inhabitant dermatophytes have infrequently been focused on the impact of the soil pH, place of sampling (availability of keratin materials) and, also, seasonality existence on the recovery rate of distinct dermatophyte species (Table 3). Some studies pointed out that soils with nutrients at alkaline pH (range 7.0–8.0) enhance the keratinolytic activity and survival of dermatophytes [7,10,14,23,27,28]. In the studies of Pakshir et al. and Rezaei-Matehkolaei et al., from Shiraz and Ahvaz, *N. fulva* and *N. gypsea* isolates were recovered from soils with a pH range of 7.0–8.0 [10,14]. In agreement with these findings, 254 out of the 256 isolates identified in the present survey, were recovered from soils with acidity ranging from 7.1 to 9.0 (Table 1). Nevertheless, among the two most abundant species, the recovery rate of *N. gypsea* (82.3%) in soils with pH 7.1–8.0 was higher than that of *N. fulva* (54.2%), which was statistically significant. While *A. uncinatum* is also a geophilic species known to be most frequently isolated from humus-rich soils, the infrequency of this species in the current survey can be attributed to the low degree of acidity in most soils samples from Iran (pH ≥ 7). Statistically substantiated studies from Europe [21,22,24,25] have reported that *A. uncinatum* in an exceptionally acidophilic species that is less-abundantly isolated from soils with pH > 6.0. In different inquiries from Iran [10,12,14], Tunisia [29] India [1,2,30], Saudi Arabia [7], the USA [18] and Brazil [27], the dermatophytes were isolated from a wide variety of soils, especially from shady and wet places rich in organic substances, e.g., gardens, public grasslands, sludge and cultivated fields, or locations with a high animal keratin content. Conversely, salty soils, waterless and beach sands and soil specimens from fruitless roadsides proved to be poor resorts for keratinophilic fungi. Concomitant with these facts, in our study, the source of the soil sample had a statistically significant effect (*p* < 0.05) on the isolation of distinct species, so that *N. fulva* was more isolated from animal habitats, parks and gardens, while *N. gypsea* was more recovered from the soils of home ranges and schools. In view of the geographic distribution, as illustrated in Figure 1, the isolation rate of the dermatophytes was evidently higher in Mazandaran than in the other provinces. In addition to *N. fulva*, the maximum isolation rate of *N. gypsea* from the soils of this province (91% of all the isolates) was also geo-statistically significant.

In view of the seasonality dependence, there are very rare data on whether or not dermatophytes have seasonal patterns in their soil occurrences. In our previous study from Ahvaz, southwest of Iran, the dermatophytes were statistically more recovered from soils collected in the autumn than in the spring (*p* < 0.05). Ahvaz, the capital of Khuzestan, is known for its semi-desert climate different from most parts of the country, with long and hot summers and short winters [10]. In the present study, the extension of the assessment to cities from 11 additional provinces with different climatic conditions led to a statistically significant difference (*p* < 0.05) in the recovery rate of the two dominant species, meaning was more frequently isolated in spring (85.3%), while *N. fulva* was equally detected whatever the season. Then, the isolation of *N. gypsea* was shown to be affected by both the location and climate. The isolation of *A. quadrifidum*, *A. tuberculatum* and *A. gertleri* had less clinical significance, but their isolation had epidemiological implications, because the species were reported for the first time in Iran.

## 5. Conclusions

In conclusion, geophilic dermatophytes do occur in various soils from different parts and climates of Iran. The predominance of *N*. *fulva* over the others was a noticeable finding, and the species is likely the main dermatophyte species recycling of keratinous materials in the soil of the country. All the species observed in this study have been incriminated in human and animal infections and, thus, have to be accurately separated from other nonpathogenic fungi. For the definite identification of soil-inhabitant dermatophytes, DNA-based identification is strongly recommended.

## Figures and Tables

**Figure 1 jof-07-00345-f001:**
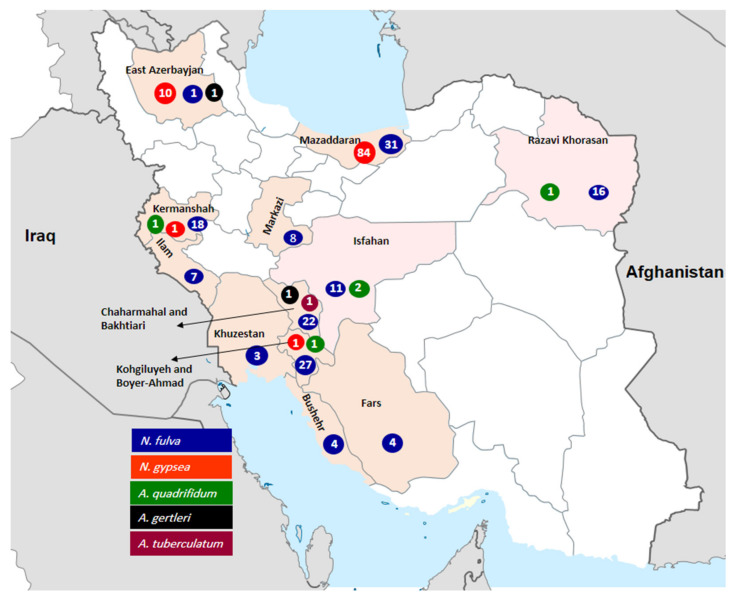
Frequency and distribution of isolated geophilic species according to the geographic location.

**Figure 2 jof-07-00345-f002:**
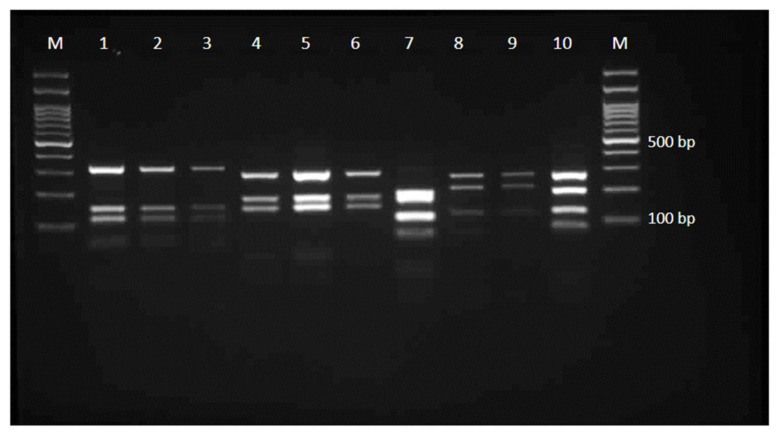
Electrophoretic profiles obtained with representative dermatophyte isolates for ITS-rDNA digested with MvaI. Lanes 1–3 = *N. fulva*, 4–6 = *N. gypsea*, 7 = *A. tuberculatum*, 8–9 = *A. gertleri* and 10 = *A. quadrifidum*.

**Table 1 jof-07-00345-t001:** Distribution of the geophilic isolates regarding the sources and pH of the soils.

Species	Soil Source						pH Range			Season				Total
	Animal Habitats	Park and Garden	School and University	Home Range	Riverside	Mountain and Roadside	6–7	7.1–8	8.1–9	Spring	Summer	Autumn	Winter	
*N. fulva*	78	75	27	21	4	9	2	116	96	96	60	34	24	214
*N. gypsea*	4	4	17	9	-	-	-	28	6	29	1	4	0	34
*A. quadrifidum*	3	2	-	-	-	-	-	4	1					5
*A. gertleri*	2	-	-	-	-	-	-	-	2	0	0	2	0	2
*A. tuberculatum*	-	1	-	-	-	-	-	-	1	1	0	0	0	1
Total	87	82	44	30	4	9	2	148	106	129	67	40	24	256

**Table 2 jof-07-00345-t002:** ITS-RFLP profiles of the keratinophilic fungi identified in this study.

Species	Size of ITS-rDNA	Size of Digested ITS-rDNA	GenBank Accession No.
*N. fulva*	652	322, 147, 112, 52, 19	MG572978-MG573055
*N. gypsea*	666	289, 179, 146, 33, 19	MG573057-MG573059
*A. gertleri*	655–656	268, 212, 117, 59	MG561646-MG561647
*A. quadrifidum*	661	268, 196, 121, 76	MG561441-MG561442
*A. tuberculatum*	677	198, 175, 114, 106, 62, 22	MT573332

**Table 3 jof-07-00345-t003:** A summary on the occurrence of dermatophytes in the soils from various countries.

Reference	Country (Year)	Soil pH with the Highest Isolation	Source with the Most Positivity Rate	Identification Method	Diversity of Recovered Species	The Dominant Isolated Species
Pakshir et al. [14]	Iran (2013)	7.0–9.0	Parks	ITS sequencing	*N. gypsea, N. fulva*	*N. gypsea*
Rezaei-Matehkolaei et al. [10]	Iran (2017)	7.0–7.9	Under trees	ITS sequencing	*N. fulva, M. canis, T. mentagrophytes*	*N. fulva*
Dehghan et al. [12]	Iran (2019)	ND *	Parks	ITS sequencing	*N. fulva, T. mentagrophytes*	*N. fulva*
Balajee et al. [1]	India (1997)	7.0–7.5	Garden and park	Mating test	*N. gypsea, N. fulva, T. mentagrophytes*	*N. gypsea*
Sharma et al. [17]	India (2008)	ND	Public places	ITS sequencing	*N. persicolor, N. fulva, N. gypsea*	*N. persicolor*
Rizwana et al. [7]	Saudi Arabia (2012)	ND	Garden	Morphology	*N. gypsea, M. canis, T. rubrum, T. mentagrophytes*	*N. gypsea*
Giugnani et al. [18]	USA (2020)	ND	Cultivated fields	Morphology	*N. fulva, N. gypsea*	*N. fulva*
Gugnani et al. [19]	St. Kitts and Nevis (2012)	ND	Under trees	Morphology	*N. gypsea, N. fulva*	*N. fulva*
Taha et al. [20]	Egypt (2018)	ND	Roadside	ITS sequencing	*N. gypsea, T. mentagrophytes, N. fulva, T. benhamiae, A. multifidum*	*N. gypsea*
Pontes et al. [21]	Brazil (2008)	7.0–8.0	Slum	Morphology	*T. terrestre, T. mentagrophytes, T. verrucosum, T. tonsurans, N. gypsea*	*T. mentagrophytes*
Jain et al. [2]	India (2011)	7.0–8.0	Roadside and garden	Morphology	*T. rubrum, T. simii, T. mentagrophytes, T. terrestre, T. verrucosum, N. fulva, M. canis, M. audouinii, E. floccosum*	*T. mentagrophytes*
Anane et al. [22]	Tunisia (2015)	ND	Animal habitats	Morphology	*N. gypsea, A. cuniculi, A. curreyi*	*N. gypsea*
Kačinová et al. [23]	Austria (2013)	7.0–8.0	Animal habitats	Morphology	*A. uncinatum* (*T. ajelloi*)*, N. gypsea, T. terrestre*	*A. uncinatum*
Javoreková et al. [24]	Slovakia (2012)	5.0–6.0	National parks	Morphology	*A. uncinatum, A. multifidum, Microsporum *sp.,* T. terrestre*	*A. uncinatum*
Ciesielska et al. [22]	Poland (2014)	3.0–5.0	ND	ITS-RFLP	*A. uncinatum*	*A. uncinatum*
Caretta et al. [21]	Italy (1992)	ND	Parks	Morphology	*A. uncinatum, N. gypsea*	*A. uncinatum*
Bohacz et al. [25]	Poland (2012)	3.4–4.4	Arable fields	Morphology	*A. uncinatum*	*A. uncinatum*

* ND = not determined.

## Data Availability

All relevant data are within the manuscript. More supplementary data are available on request.

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
