# Peer review of "Diversity of Geophilic Dermatophytes Species in the Soils of Iran; The Significant Preponderance of *Nannizzia fulva"

_jof, 2021, doi:10.3390/jof7050345_

Round 1
Reviewer 1 Report
Dear Authors,
The manuscript entitled “Diversity of geophilic dermatophytes species in the soils of Iran; the significant preponderance of Nannizzia fulva” is well written and structured. The study provides interesting information about the occurrence of different geophilic dermatophytes in various soils from different parts and climates in Iran. I think it would be worthwhile to perform such studies in several parts of the world to have an overview of the epidemiology of this specific group of dermatophytes.
I have only some minor comments that could eventually be addressed in the discussion:
- Do you have any idea about the potential of MALDI-TOF MS in the identification of this specific group of dermatophytes? Can they be separated via this methodology? Although the used DNA-based identification technology appears to be effective for identification up to species level it seems, in my opinion, quite time consuming and expensive in comparison with the Maldi identification tool.
- Additionally, did you perform some antifungal susceptibility tests on the strains? Do you have any idea about their profile or known resistance in this specific group of geophilic dermatophtyes?
Author Response
Reviewer #1 comment
Dear Authors,
The manuscript entitled “Diversity of geophilic dermatophytes species in the soils of Iran; the significant preponderance of Nannizzia fulva” is well written and structured. The study provides interesting information about the occurrence of different geophilic dermatophytes in various soils from different parts and climates in Iran. I think it would be worthwhile to perform such studies in several parts of the world to have an overview of the epidemiology of this specific group of dermatophytes.
I have only some minor comments that could eventually be addressed in the discussion:
Authors' response: the authors thank the respect reviewer on his/her positive sense for our paper.
Comment: Do you have any idea about the potential of MALDI-TOF MS in the identification of this specific group of dermatophytes? Can they be separated via this methodology? Although the used DNA-based identification technology appears to be effective for identification up to species level it seems, in my opinion, quite time consuming and expensive in comparison with the Maldi identification tool.
Authors' response: To be honest, I have no knowledge or experience on MALDI-TOF MS for identification of dermatophytes. But I confirm that there are good agreement of 99 %-100% in the identifications obtained by PCR-sequencing and MALDI-TOF MS techniques (Nenoff et al. Medical mycology. 2013;51(1):17-24. - Packeu et al. Journal of clinical microbiology. 2014;52(9):3440-3) and it can be done by developing a library of MALDI-TOF MS with standard and local geophilic dermatophyte species/isolates. Nevertheless, given to the high number of geophilic species and high degree of intra-species variation in geophilic species identification of these fungi by MALDI-TOF MS can be interesting and the matter of research for next studies.
Comment: Additionally, did you perform some antifungal susceptibility tests on the strains? Do you have any idea about their profile or known resistance in this specific group of geophilic dermatophtyes?
Authors' response: The antifungal susceptibility of some of these isolates is the matter of another study that currently is being done. Till now, we had no evidence of resistance (terbinafine or Itraconazole) among tested isolates and generally antifungal is not popular among geophilic species which are known to make acute dermatophytosis. Recently, we documented confirmed cases of terbinafine resistance among some clinical Trichophyton isolates (T. indotinea). However, to the best of our knowledge, no evidence of such resistance among Microsporum, Nannizzia or Arthroderma isolates have yet been recorded been from Iran.
Reviewer 2 Report
Interesting topic and results. However, the discussion needs to be significantly redrafted as it is very regionally oriented. Moreover, I propose to present the results in a more synthetic way. The current form is too extensive and presents the same information several times. In several places, attention must be paid to text formatting.
Detailed notes on each section below:
- Abstract:
line 36: incorrectly formatted text - ribosomal DNA
line 39: the term pH is misspelled
- Introduction:
line 52-53: de Hoog et a. (Mycopathologia 2017 Feb;182(1-2):5-31.doi: 10.1007/s11046-016-0073-9) lists 9 types of molecularly identified dermatophytes and the authors give only 7. It should be explained why.
line 61-62: incorrectly formatted text, font size is not uniform
line 79: please specify the purpose of the study more precisely. The term DNA-sequence based methods is too general.
- Methods:
Paragraphs 2.1 and 2.2 can be combined into one as the location and selection of sites for collection of material for analysis. Also, the abbreviation gr is invalid for grams. Rather, g is used. The equipment and the manufacturer of the pH meter should also be listed.
Paragraph 2.6 - please specify the program in which the statistical analysis was performed.
- Results
I propose to delete Tables 2 as Fig. 1 contains the same information. The accession numbers of the sequences can be listed in Table 1. In addition, the data from Table 3 can also be configured with Table 1. The layout of rows and columns is the same, and the last column, total, contains the same data as in Table 1.
In Fig. 2 the top photo is unnecessary, it is enough to mention the size of the amplification product in the text. A photo with an electrophoretic profile after digestion can be enlarged and more accurately described the sizes of products obtained after the action of the restriction enzyme.
- Discussion
There are also data from similar studies done in Europe. I propose to supplement the discussion by comparing the authors' own results with these reports. In the current version, the discussion is exclusively regional.
Moreover, I propose to shorten the discussion by removing geographic information from it. It seems reasonable to limit them to a minimum, as they are not of interest to an international group of journal readers.
I propose to delete the fragment on lines 101-111, it has nothing to do with the purpose of the article.
Author Response
Reviewer #2
Interesting topic and results. However, the discussion needs to be significantly redrafted as it is very regionally oriented. Moreover, I propose to present the results in a more synthetic way. The current form is too extensive and presents the same information several times. In several places, attention must be paid to text formatting.
Detailed notes on each section below:
Authors' response:
We would like to thank the respect reviewer for his/her helpful and detailed comments. We did our best to improve the manuscript. Below are our responses to the comments.
- Abstract:
Comment: line 36: incorrectly formatted text - ribosomal DNA
Authors' response: the reviewer is right. We adjusted the text size and format.
Comment: line 39: the term pH is misspelled
Authors' response: the spell was corrected in all parts of the manuscript.
- Introduction:
Comment: line 52-53: de Hoog et a. (Mycopathologia 2017 Feb;182(1-2):5-31.doi: 10.1007/s11046-016-0073-9) lists 9 types of molecularly identified dermatophytes and the authors give only 7. It should be explained why.
Authors' response: we thank the respect reviewer for this interesting question. Regarding your questions, I asked some colleagues who like me were the co-authors of the paper by de Hoog et al. and they believe that currently 7 genera are classified in the dermatophytes, the family Arthrodermataceae. Guarromyces and Ctenomyces are known as dermatophyte-like (dermatophytoid) fungi. Guarromyces is classified in a different family, the Onygenace as well as Ctenomyces which belongs to the Gymnoascaceae. In Fig. 3 (page 12) of the mentioned paper by de Hoog et al., Guarromyces was selected as out-group.
Comment: line 61-62: incorrectly formatted text, font size is not uniform.
Authors' response: the font and text size were unified.
Comment: line 79: please specify the purpose of the study more precisely. The term DNA-sequence based methods is too general.
Authors' response: we meant PCR-RFLP and PCR-sequencing from the term DNA-sequence based and in this regard we rephrased the mentioned sentence and now we think it make sense.
- Methods:
Comment: Paragraphs 2.1 and 2.2 can be combined into one as the location and selection of sites for collection of material for analysis. Also, the abbreviation gr is invalid for grams. Rather, g is used. The equipment and the manufacturer of the pH meter should also be listed.
Authors' response: We combined two paragraph and the abbreviation was correct. The features of the used pH meter was addressed as (Knick Portamess® 911 pH meter, Berlin, Germany).
Comment: Paragraph 2.6 - please specify the program in which the statistical analysis was performed.
Authors' response: We use SPSS software version 21 (IBM, USA) for our analysis and specify it in the revised paper.
- Results
Comment: I propose to delete Tables 2 as Fig. 1 contains the same information. The accession numbers of the sequences can be listed in Table 1. In addition, the data from Table 3 can also be configured with Table 1. The layout of rows and columns is the same, and the last column, total, contains the same data as in Table 1.
Authors' response: We agree that Fig. 1 have some data of Table 2 and as the respect reviewer suggested, we deleted Tables 2 & 3 and combined the data of Table 3 with Table 1. Besides, we provided a new Table 2 containing accession numbers and the exact sizes of PCR-RFLP fragments.
Comment: In Fig. 2 the top photo is unnecessary, it is enough to mention the size of the amplification product in the text. A photo with an electrophoretic profile after digestion can be enlarged and more accurately described the sizes of products obtained after the action of the restriction enzyme.
Authors' response: We deleted the top photo for PCR products.
- Discussion
Comment: There are also data from similar studies done in Europe. I propose to supplement the discussion by comparing the authors' own results with these reports. In the current version, the discussion is exclusively regional.
Authors' response: We thank the reviewers for such comments. I wish he/she could address some similar studies from Europe for our use because such investigations exactly compatible to ours are limited and very old. I even asked some European colleagues to send me such studies but they could not find helpful documents for my purpose. Nevertheless, I found 6 studies from Italy, Poland, Austria and Slovakia and used them to improve the Discussion and added two paragraphs to the paper. Here I have to mention an important point. During the revise of my manuscript I found that the Table 4 (Table 3 in revised version) was damaged, incomplete and only contained summary of three studies from Iran and I guess that's why the reviewer thought the Discussion part of our manuscript is extensively national (regional). I contacted Ms. Caroline Song, Assistant Editor of Journal of Fungi for this issue and she replied me that they made no change on the table and the original table uploaded to the journal was as this. I don't know what actually happened. But the original table included the review of 12 studied from Asia, Africa and America. In the revised paper the table (Table 3) and Discussion were supplemented with all 12 previous reviewed studies and some new studies from Europe.
Comment: Moreover, I propose to shorten the discussion by removing geographic information from it. It seems reasonable to limit them to a minimum, as they are not of interest to an international group of journal readers.
I propose to delete the fragment on lines 101-111, it has nothing to do with the purpose of the article.
Authors' response: The mentioned fragment (lines 101-111) was removed from the discussion. We also removed some unnecessary sentences in line with the reviewer comment to shorten the Discussion.
Round 2
Reviewer 2 Report
Interesting article. Congratulations!